# Lifetime estimate for plasma turbulence

Yasuhito Narita[1,2,3] and Zoltán Vörös[1,2,4]

[1] Space Research Institute, Austrian Academy of Sciences, Schmiedlstr. 6, A-8042 Graz, Austria
[2] Institute of Physics, University of Graz, Universitätsplatz 5, A-8010 Graz, Austria
[3] Institut für Geophysik und extraterrestrische Physik, Technische Universität Braunschweig, Mendelssohnstr. 3, D-38106 Braunschweig, Germany
[4] Geodetic and Geophysical Institute, RCAES, Hungarian Academy of Sciences, Sopron, Hungary

*Correspondence to:* Y. Narita
(yasuhito.narita@oeaw.ac.at)

**Abstract.** A method is proposed to experimentally determine the intrinsic time scale or a decay rate of turbulent fluctuations. The method is based on the assumption that the Breit-Wigner spectrum model with a non-Gaussian frequency broadening is valid in the data analysis. The decay rate estimate is applied to the multi-spacecraft magnetic field data in interplanetary space, **yielding the result** that the decay rate on spatial scales of about 1000 km (about 10 times larger than the ion inertial length) **which** is higher than the theoretical predictions from the random sweeping time scale of the eddy turnover time. The faster decay of fluctuation components in interplanetary space is interpreted as a realization of plasma physical (and not fluid mechanical) **processes**.

## 1 Introduction

Turbulent fluctuations appear in various fluid or gaseous media, and exhibit both temporally developing and spatially varying fields. The fluctuation properties are characterized by correlations between two spatial points and between two different times. **A Fourier representation** of the two-time, two-point correlation is the energy spectrum in the wavenumber-frequency domain.

**Our experimental access** to the wavenumber-frequency spectrum is limited. One can easily obtain the frequency spectra from time series data using a single probe, whereas one has to determine the wavenumber spectra using multiple probes simultaneously. Moreover, the Fourier transform cannot be performed if the number of probes is small (say, below 10). For this reason, Taylor's frozen-in flow hypothesis (Taylor, 1938) is widely implemented to the measurement of the turbulence energy spectra. Taylor's hypothesis assumes that fluctuating fields are "frozen-in" into the high-speed flow such that the time series data of fluctuations are interpreted as spatial structures swept by the flow and passing by the probe. In other words, the spectral energy is assumed to lie on the Doppler shift, $\omega = k_\mathrm{s} U_0$, where $\omega$ denotes the frequency in the Eulerian frame, $k_\mathrm{s}$ the streamwise wavenumber (the wavenumber in the direction to the flow), and $U_0$ the mean flow speed.

Time evolution is neglected during the time period of measurements when one uses Taylor's hypothesis, so the intrinsic time scale information cannot be estimated from single-point measurements. Of course, Taylor's hypothesis breaks down whenever the frozen-in condition of the fluctuating field is violated, for example, when the flow speed has large-scale variations (the ran-

dom sweeping model), or when propagating waves exist in the turbulent fluctuations. Is it then ever possible to experimentally determine the intrinsic time scale in turbulence studies?

Here we develop a method to experimentally determine the lifetime of fluctuation components in plasma turbulence. The task of measuring the wavenumber-frequency spectrum is achieved by using multi-point magnetic field measurements by Cluster spacecraft in interplanetary space (Escoubet et al., 2001) and a high-resolution interferometric spectral estimator (Narita et al., 2011). The spatial scales of consideration is 1000 km (about 10 times larger than the ion inertial length). The broadening of the energy spectrum around the Doppler shifted frequencies (sideband formation) is interpreted as a realization of temporally decaying waves in turbulence. **For the goal of** measuring the lifetime, the concept of Breit-Wigner spectrum is introduced and applied to the spacecraft data. The measure of the frequency broadening around the Doppler shift also serves as a test for Taylor's frozen-in hypothesis in the spectral domain (**Taylor, 1938**) .

**It is worthwhile to note that other choices are possible for analytically expressing or fitting the non-Gaussian shape of the frequency-sliced spectrum (a slice of the energy spectrum over the frequencies at a fixed value of the wavenumber). The kappa distribution is a likely candidate. Underlying physical models are different between the kappa distribution and the Breit-Wigner distribution. The use of kappa distribution is developed for describing non-extensive statistical mechanics (e.g., non-extensive entropy), and is developed for discrete particles that have long-range interactions and correlations. Its application to the energy spectra for continuous turbulent fields is still in question, in particular, in interpreting the control parameter kappa in the distribution . On the other hand, the use of a Breit-Wigner (or Lorentzian) distribution has a solid background with a physical model in that the decay rate (appearing as an imaginary part of the frequency) can be measured experimentally and then immediately compared to wave or turbulence models.**

## 2   Breit-Wigner spectrum

One may extend the frequencies **from the real numbers (as oscillatory part) to the complex numbers** by including an imaginary part for a temporal damping. We combine the excitation frequency $\omega_0$ and the decay rate $\gamma$ as

$$\omega_0 \to \omega_0 - \mathrm{i}\frac{\gamma}{2} \tag{1}$$

The wave field $\Phi$ (e.g., flow velocity, density, or magnetic field) is expressed as

$$\Phi(t) = \Phi(0)\exp\left[-\mathrm{i}\left(\omega_0 - \frac{\mathrm{i}\gamma}{2}\right)t\right] \tag{2}$$

where $\Phi(0)$ is the initial wave amplitude. The wave energy is estimated by the square amplitude:

$$P(t) = |\Phi(t)|^2 = |\Phi(0)|^2\exp\left[-\gamma t\right]. \tag{3}$$

The lifetime of wave excitation $\tau$ is an inverse of the decay rate

$$\tau = \frac{1}{\gamma}. \tag{4}$$

The decay rate is associated with the frequency broadening around the peak frequency in the Breit-Wigner spectrum.

The picture under consideration is that turbulent fluctuations are decaying while being excited intermittently or continuously at different wavenumbers to sustain the energy cascade balance. It is assumed that the fluctuations are excited and subject to decay with a rate equivalent to the frequency broadening $\gamma$.

The Fourier transform of the wave field (which turns out to be the same as the Laplace transform with respect to the decay rate $\gamma$) **is**:

$$\tilde{\Phi}(\omega) \quad = \quad \frac{1}{\sqrt{2\pi}} \int_0^\infty \Phi(t) \exp\left[\mathrm{i}\omega t\right] \mathrm{d}t \tag{5}$$

$$= \quad \frac{1}{\sqrt{2\pi}} \int_0^\infty \Phi(0) \exp\left[\left(\mathrm{i}(\omega_0 - \omega) + \frac{\gamma}{2}\right)t\right] \mathrm{d}t \tag{6}$$

$$= \quad \frac{1}{\sqrt{2\pi}} \Phi(0) \frac{1}{\mathrm{i}(\omega_0 - \omega) + \gamma/2}. \tag{7}$$

The frequency broadening is a measure of wave lifetime. According to the Breit-Wigner formula in nuclear resonance phenomena of the compound-state formation (Breit and Wigner, 1936; Bohr, 1936),

$$P(\omega) \quad = \quad |\tilde{\Phi}(\omega)|^2 \tag{8}$$

$$= \quad \frac{1}{2\pi} |\Phi(0)|^2 \frac{1}{(\omega_0 - \omega)^2 + (\gamma/2)^2} \tag{9}$$

$$= \quad \frac{1}{\gamma} |\Phi(0)|^2 f_{\mathrm{BW}}(\omega, \gamma) \tag{10}$$

where

$$f_{\mathrm{BW}}(\omega, \gamma) = \frac{1}{\pi} \frac{\gamma/2}{(\omega_0 - \omega)^2 + (\gamma/2)^2} \tag{11}$$

is the Breit-Wigner distribution widely applied in the field of nuclear resonance phenomena. The Breit-Wigner distribution has a higher probability than that of a Gaussian distribution at a position or a variable value $x$ larger than the standard deviation of the distribution (Fig. 1). The peak of $f_{\mathrm{BW}}$ at the resonant frequency $\omega_0$ is given as $f_{\mathrm{BW}} = \frac{2}{\pi\gamma}$, **so that the** half-value of the peak is $\frac{1}{\pi\gamma}$, and **the** half-value width is realized when the decay rate $\gamma$ satisfies the condition

$$\frac{1}{\pi\gamma} = \frac{1}{\pi} \frac{\gamma/2}{(\Delta\omega/2)^2 + (\gamma/2)^2}, \tag{12}$$

namely,

$$\Delta\omega = \gamma. \tag{13}$$

Thus, the decay rate can be estimated by measuring the half-value-width frequency of the spectral peak. The decay rate $\gamma$ is directly associated with the lifetime of the excited wave component as $\tau = \gamma^{-1}$, and justifies Eq. (4).

The shape of the Breit-Wigner spectrum depends on two parameters: the spectral peak frequency $\omega_0$ and the half-value-width (or the decay rate) $\gamma$. A fitting procedure is needed to estimate the two parameters from the measured spectrum. In this work, the Levenberg-Marquardt algorithm for the least square fitting (Levenberg, 1944; Marquardt, 1963) is used with a detailed numerical implementation in Press (1992).

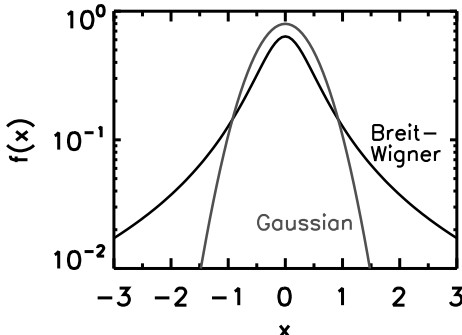

**Figure 1.** Breit-Wigner distribution for a half-value-width of 0.5 ($\omega_0 = 0, \omega \to x, \gamma/2 = 0.5$ in Eq. (11)) in black and Gaussian distribution with a standard deviation of $\sigma = 0.5$ (in gray).

## 3 Application to space plasma

**Observational setup**

The Breit-Wigner spectrum is tested against turbulent fluctuations in interplanetary space. Four Cluster spacecraft (Escoubet et al., 2001) encounter the solar wind ahead of the Earth bow shock. The observational setup is illustrated in Fig. 2 for the
time interval on 17 March 2005, 1000–1200 UT. The Cluster spacecraft are in the solar wind and form a nearly regular tetrahedron with an inter-spacecraft distance of about 1000 km. For reference, the ion inertial length is on the order of 100 km in the solar wind. The solar wind plasma streams radially away from the sun at a mean speed of about 373 km/s, measured by the electrostatic ion analyzer on board the Cluster (Rème et al., 2001). The interplanetary magnetic field is almost in the plane spanning the sunward and the duskward directions, and is highly inclined dawnward. The mean magnetic field is:
$B_{\mathrm{x}} = 2.8\,\mathrm{nT}$, $B_{\mathrm{y}} = -4.2\,\mathrm{nT}$, and $B_{\mathrm{z}} = 0.6\,\mathrm{nT}$ in the GSE (geocentric solar ecliptic) coordinate system. The mean ion number density is about 13 cm$^{-3}$. The ion temperature is 0.18 MK and 0.15 MK parallel and perpendicular to the mean magnetic field, respectively. The observed time interval represents a low-speed plasma stream of the solar wind, and exhibits a large-scale pressure-balanced structure between the magnetic pressure and the ion thermal pressure, indicated by an anti-correlation of the field variations between the magnetic field magnitude and the ion number density in Fig. 3. **The value of ion beta is about 1.5,**
**the Alfvén speed 41 km s$^{-1}$, the ion gyro-frequency 0.66 rad s$^{-1}$ (for protons), the ion inertial length 63 km rad$^{-1}$. The highest wavenumber resolved in the data analysis is $kV_{\mathrm{A}}/\Omega_{\mathrm{i}} = 0.16$, where $k$ is the wavenumber, $V_{\mathrm{A}}$ the Alfvén speed, and $\Omega_{\mathrm{i}}$ the ion gyro-frequency, respectively. The Cluster tetrahedral configuration during the analyzed time interval represents transition spatial scales from the magnetohydrodynamic to the ion-kinetic range. The gyro-radius for the thermal protons is estimated as about 84.6 km rad$^{-1}$.** The observed time interval is suited to testing for the Breit-Wigner
spectrum for the nearly constant mean plasma flow speed over the interval.

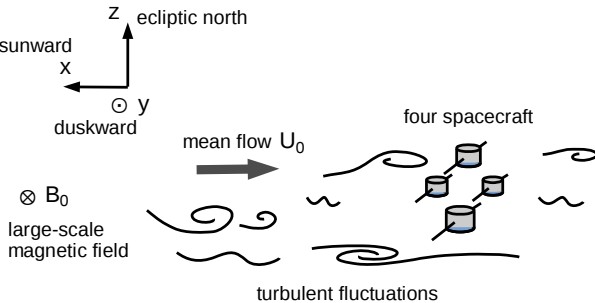

**Figure 2.** Sketch of the observational setup.

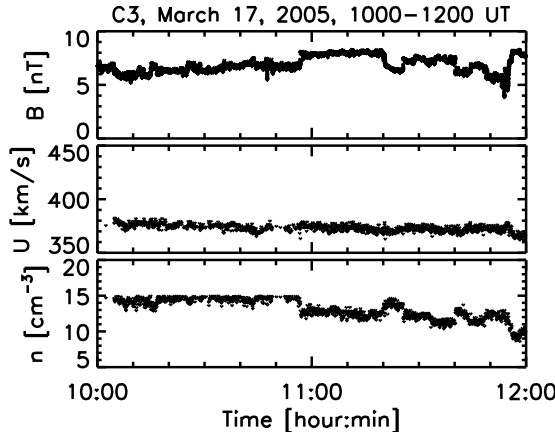

**Figure 3.** Time series data in the solar wind.

### Streamwise wavenumber-frequency spectrum

The streamwise wavenumber-frequency spectrum for the magnetic field fluctuations is displayed in Fig. 4. The spectrum exhibits an extended structure from the origin of the spectral domain (zero wavenumber and zero frequency) to **a wavenumber of 0.002 rad km$^{-1}$ and an angular frequency of about 0.75 rad s$^{-1}$, indicating a phase speed of about 375 km s$^{-1}$.** The spectral extension is almost linear, and the slope of the extension (the propagation speed estimated by from the ratio of the frequency to the wavenumber) roughly agrees with the mean flow speed, **373 km s$^{-1}$.** The spectral extension shows a broadening, which can be interpreted either as a broadening over the wavenumbers or that over the frequencies. We perform the data analysis by interpreting that the spectral broadening appears along the frequency axis and the broadening is a function **of wavenumber**.

**The streamwise wavenumber-frequency spectrum is constructed as follows. First, we collect the magnetic field data from each spacecraft on the analyzed time interval and Fourier-transform the magnetic field fluctuations (after sub-**

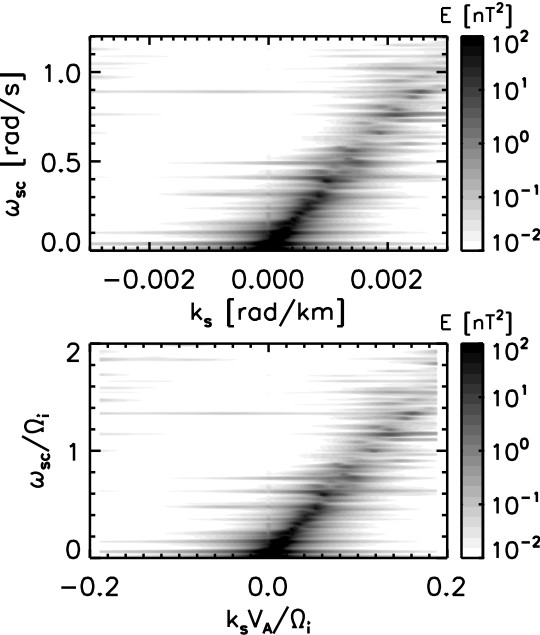

**Figure 4.** Eulerian (spacecraft-frame) energy spectrum for magnetic field fluctuations in the solar wind in the streamwise wavenumber-frequency domain. The measurements of Cluster spacecraft on a time interval of 17 March, 2005, 1000–1200 UT are used. **The spectral are displayed in physical units (rad km$^{-1}$ for the wavenumber and rad s$^{-1}$ for the frequency) in the upper panel, and in normalized units using the ion inertial length $V_A/\Omega_i = 63$ km rad$^{-1}$ and the ion gyro-frequency $\Omega_i = 0.66$ rad s$^{-1}$.**

**tracting the constant, mean field part) from the time domain into the spacecraft-frame frequencies. The four frequency-domain magnetic field data are put together into a twelve-by-twelve covariance matrix (consisting of three magnetic field components from each spacecraft). The covariance matrix is projected onto the streamwise wavenumbers at each frequency using the wave telescope technique with an improvement with the MSR technique (Multi-point Signal Res-**
5   **onator) by making use of the orthogonality of the noise eigenvectors and the signal eigenvectors (Motschmann et al., 1996; Glassmeier et al., 2001; Narita et al., 2011). The projected matrix is a three-by-three spectral density matrix as a function of frequency and streamwise wavenumber. Finally, the energy is estimated by taking the trace of the matrix. Five-hertz down-sampled magnetic field data from all the four Cluster spacecraft are used in the analysis. The frequencies and the wavenumbers can also be normalized to the ion gyro-frequency $\Omega_i$ (for protons) and the ion inertial length**
10   **$V_A/\Omega_i$, respectively (Fig. 4).**

**Fitting to the Breit-Wigner spectrum**

The wavenumber-frequency spectrum is obtained for the magnetic field data of Cluster (Balogh et al., 2001) using the wave telescope projection technique with an MSR extension (multi-point signal resonator) to eigenvalue-based projection (Narita et al., 2011). The spectrum is displayed in Fig. 5, and is analyzed to determine the peak frequency and the broadening frequency

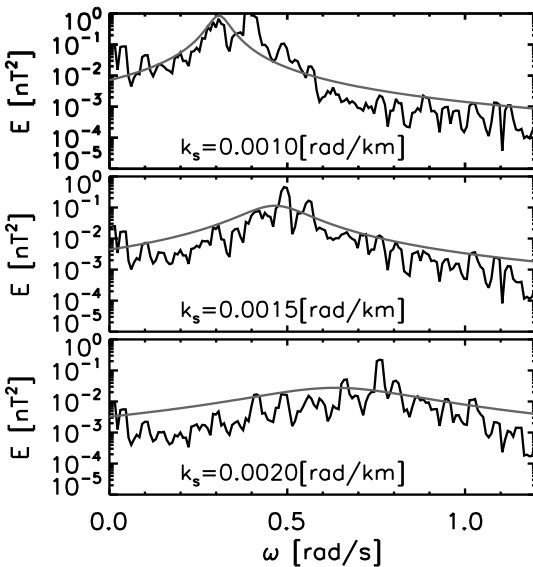

**Figure 5.** One-dimensional slice of the energy spectrum over the spacecraft-frame frequencies at a streamwise wavenumber of 0.015 [rad/km] from Fig. 4 (in black) and a fitted curve by the Breit-Wigner distribution (in gray) for $\omega_0 = 0.49$ [rad/s] and $\gamma = 0.08$ [rad/s].

using the Breit-Wigner form of spectrum. To perform the fitting, the energy spectrum is sliced over the frequencies at a given value of streamwise wavenumber. Three examples of the frequency slice are displayed as one-dimensional spectra in Fig. 5 (curved lines in black) at wavenumbers of 0.0010 rad/km, 0.0015 rad/km, and 0.0020 rad/km.

For the least-square fitting procedure, the Levenberg-Marquardt algorithm is applied to each frequency slice on the logarith-
mic scale of the spectral energy. The one-dimensional slices of the spectra and a fluctuation level of the spectra are used as inputs to the least-square fitting. **A fluctuation amplitude** of 0.2 nT$^2$ is used in the least-square fitting, which comes from the measurement of the fluctuation level of the spectrum. The peak frequency $\omega_0$ (at which the spectrum maximizes) and the broadening frequency $\gamma$ (which is a measure of the decay rate) are obtained as fitting outputs. We obtain, for example, $(\omega_0, \gamma) = (0.308 \pm 0.010\,[\mathrm{rad/s}], 0.056 \pm 0.002\,\mathrm{rad/s})$ at a wavenumber of 0.0010 rad/km, $(\omega_0, \gamma) = (0.465 \pm 0.010\,[\mathrm{rad/s}], 0.186 \pm 0.003\,\mathrm{rad/s})$
at a wavenumber of 0.0015 rad/km, and $(\omega_0, \gamma) = (0.635 \pm 0.020\,[\mathrm{rad/s}], 0.463 \pm 0.002\,\mathrm{rad/s})$ at a wavenumber of 0.0020 rad/km, respectively. The best-fitted spectra are over-plotted in Fig. 5 in gray curves.

The increasing sense of the peak frequencies and the half-value-widths at larger wavenumbers are quantitatively tracked by repeating the fitting procedure at various wavenumbers (Fig. 6). The increasing sense of the peak frequencies reasonably agrees with the the Doppler shift (dotted line), represented by $\omega = k_s U_0$, which justifies the picture of ideal convection (Taylor's
hypothesis) in the lowest-order picture (talking only about the peak frequencies).

The measured half-value-widths $\gamma$ are interpreted as the decay rate associated with the respective peak frequencies, and are compared to the estimate for the random sweeping (dotted line), represented by $\Delta\omega = \sqrt{k_s^2 \delta U^2} \propto k_s$, and that for the

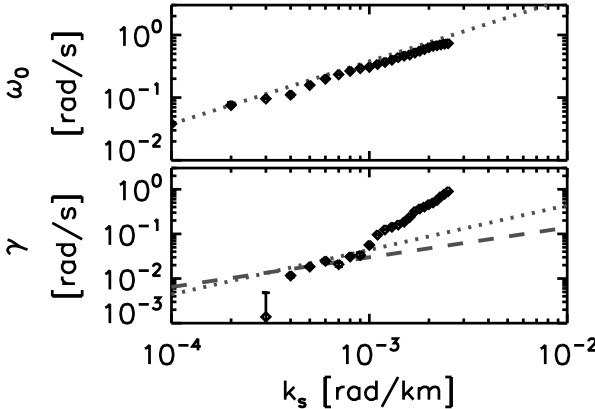

**Figure 6.** Peak frequency $\omega_0$ (upper panel) as a function of the streamwise wavenumbers with comparison to the Doppler shift (dotted line) and half-value-width $\gamma$ (lower panel) with comparison to the random sweeping (dotted line) and eddy turnover time (dashed line).

eddy turnover time (dashed line), represented by $\Delta\omega = \sqrt{k_\mathrm{s}^2 u^2} \sim \sqrt{k^2(\epsilon^{1/3}k_\mathrm{s}^{-2/3})^2} \sim \epsilon^{1/3}k_\mathrm{s}^{2/3}$. Here the value of the energy transfer rate $\epsilon$ is fitted to the measured curve of the decay rate to $\epsilon = 27$ rad km$^{-2}$ s$^{-3}$ at wavenumbers below $10^{-3}$ rad/km. At wavenumbers above $10^{-3}$ rad/km the decay rate $\gamma$ becomes increasingly larger at smaller wavenumbers. The measured increase rate is faster than the prediction from the random sweeping model and that from the eddy turnover time. In other words, the lifetime of the fluctuation components is shorter **than that of** in fluid turbulence. The increasing sense of the half-value-widths indicates that the mapping quality from the wavenumbers onto the frequencies becomes diminished at larger wavenumbers; the picture of ideal convection (Taylor's hypothesis) breaks down in the second-order picture. **The fitting errors are added in Fig. 6 as the frequency error bar and the decay rate error bar. The fitting error is the largest only at the smallest wavenumbers. The wavenumber for the gyro-radius is $k_\mathrm{gi} = 1.18 \times 10^{-2}$ rad km$^{-1}$, and is beyond the plot range in Fig. 6.**

**Dispersion relation**

**The wave frequencies are corrected for the Doppler shift using the relation $\omega_\mathrm{rest} = \omega_0 - k_\mathrm{s}U_0$, and compared with the dispersion relations for the whistler mode and the kinetic Alfvén mode, both being likely candidates to explain the linear mode components of the turbulent fluctuations in the ion-kinetic range. The analytic expression of the dispersion relation for the whistler mode (Gary, 1993):**

$$\frac{\omega}{\Omega_\mathrm{i}} = \frac{kV_\mathrm{A}}{\Omega_\mathrm{i}} \left( 1 + \frac{k_\parallel^2 V_\mathrm{A}^2}{\Omega_\mathrm{i}^2} \right)^{1/2}, \tag{14}$$

**where $k_\parallel$ denotes the parallel component of the wavevector in the direction of the mean magnetic field. The dispersion relations for the whistler mode and the kinetic Alfvén mode are computed for the measured local mean field values. A propagation angle of 85 degree is used, by referring to the statistical result of the earlier Cluster measurements**

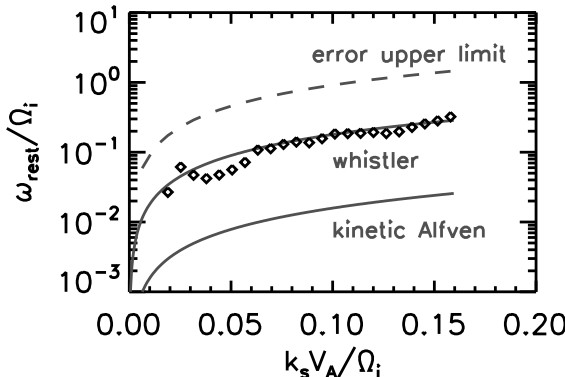

Figure 7. Estimated peak frequencies in the plasma rest frame (corrected for the Doppler shift) as a function of the streamwise wavenumbers (data points in black), upper limit of the frequency estimate error (dashed line in gray), and dispersion relations for the whistler mode (middle curve) and the kinetic Alfén mode (lower curve).

(Perschke et al., 2014). The dispersion relation for the kinetic Alfvén mode is analytically obtained from the two-fluid model of plasma (Lysak and Lotko, 1996; Hollweg, 1999; Lysak, 2008):

$$\frac{\omega}{\Omega_{\mathrm{i}}} = \frac{k_{\|}V_{\mathrm{A}}}{\Omega_{\mathrm{i}}} \left[ 1 + \frac{\beta}{2} \left( \frac{k_{\perp}V_{\mathrm{A}}}{\Omega_{\mathrm{i}}} \right)^2 \right]^{1/2} \tag{15}$$

Comparison with the dispersion relations is displayed in Fig. 7. Even though the both modes can explain the measured rest-frame frequencies, the agreement with the whistler mode is leading for the exact frequency matching in a wider range of the wavenumbers. The frequency error in the plasma rest frame mostly comes from the uncertainty in the flow velocity measurement, and only upper limits can be shown in the logarithmic plot style.

## 4   Discussion and Outlook

Decay rate is not constant over the spatial scales, but becomes increasingly larger (faster decay) toward smaller spatial scales. The increasing sense of decay at larger wavenumbers agrees with the picture of fluid turbulence, but the decay rate becomes increasingly larger than anticipated from the fluid turbulence models (random sweeping model, eddy turnover model). The larger values of the decay rate should be interpreted as a nature of collisionless plasma turbulence.

Various scenarios are possible to explain the deviation of the decay rate from the estimates based on fluid turbulence (random sweeping time or eddy turnover time). First, wave-wave interactions are in operation and generate other fluctuation types such as electric field or density fluctuations, transferring the magnetic field fluctuation energy into the other energy types. Second, wave-particle interactions are in operation, too, such as coherent scattering (Landau or cyclotron resonance) or incoherent scattering (pitch angle scattering). **Indeed, the agreement with the whistler-mode dispersion relation is indicative of the idea that the decay rate be associated with the Landau or cyclotron damping rate of the linear mode. However, it is**

**interesting that the sudden increase of the decay rate is not directly associated with the ion kinetic scales, since the wavenumber for the ion inertial length (assuming protons) is about 0.016 rad/km, and the damping rate is sufficiently small at those wavenumbers. Whether the sudden increase of the decay rate is due to a magnetohydrodynamic effect or an ion-kinetic effect remains an open question.**

To conclude the manuscript, we raise several items that should be studied more elaborately to strengthen (or disprove) the use of the Breit-Wigner spectrum in plasma turbulence research.

1. Invariance of the spectral index of one-dimensional energy spectra between the wavenumber domain $E(k)$ and the frequency domain $E(\omega)$. The invariance holds in the case of Gaussian frequency distribution. For the Breit-Wigner frequency distribution, the invariance of the spectral index is not **yet** guaranteed. Also, error estimate for Taylor's hypothesis needs to be recalculated for a non-Gaussian shape of frequency distributions. For a Gaussian frequency distribution, the error of Taylor's hypothesis can conveniently expressed by the Gauss error function.

2. The spectral peak may break up into multiple branches when different linear mode waves are excited in the turbulent field at once. The frequency slice of the spectrum will show a convolution of different branches (with different peak frequencies and different half-value-widths).

3. Another way to express the non-Gaussian shape distribution is to introduce higher-order moments. The fourth-order moment (in a non-trivial manner by measuring the deviation from a Gaussian shape, e.g., kurtosis or flatness index) is particularly suited to such a task. Perhaps there is a relation between the decay rate and the fourth-order moment of the frequency spread around the spectral peak.

*Acknowledgements.* The work conducted in Graz is financially supported by Austrian Space Applications Programme at Austrian Research Promotion Agency (FFG ASAP-12) *SOPHIE, Solar Orbiter wave observation program in the heliosphere* under contract 853994 and by Austrian Science Funds (FWF) *Twisted magnetic flux ropes in the solar wind.* under contract P28764-N27. YN acknowledges discussions with Uwe Motschmann and Karl-Heinz Glassmeier on the turbulence spectra. The work conducted in Braunschweig is financially supported by German Science Foundation under contract MO 539/20-1, *DECODE: Detection of wave coupling cascade in space plasmas.*

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
