# Peer review of "Lifetime estimate for plasma turbulence"

_Nonlinear Processes in Geophysics, 2017_

## Referee Comment (RC1) · P. Yoon (Referee) · 21 Jun 2017

Review of "Lifetime estimate for plasma turbulence" by Y. Narita and Z. Vörös

In this innovative paper the authors introduce the Breit-Wigner spectral distribution function in order to fit the observed data and extract the underlying turbulence decay rate. I believe that this is a worthwhile approach that should be made aware to the space physics community in general, since this method may be useful for unraveling some underlying physical processes. I have some questions, which I suggest the authors to consider in order to clarify the model.

(Question) From the comparative plots of spectral distributions in Figure 1, Breit-Wigner distribution appears to have a superficial similarity with the kappa distribution. It seems

to be that one may alternatively use the kappa spectral distribution to fit the data in Figure 5. So, the question is, why should one prefer BW distribution over the kappa or Lorentzian spectral function? Is there a rationale for choosing Breit-Wigner distribution over other models?

(Question) Please specify what xxx's are in the following:

Lines 19-20: wavenumber of xxx and a frequency of xxx?

Line 21: mean flow speed, xxx km/s

(Question) In Figure 4, can the angular frequency versus wave number plot be fitted with some known dispersion relations? Fast, slow, or Alfven mode?

---

## Referee Comment (RC3) · 8 Aug 2017

The authors present an interesting method, based on the Breit-Wigner model for non Gaussian frequency broadening, to look at the stream wavenumber frequency spectrum constructed from cluster measurements in the solar wind and try estimate a dispersion relation ($\omega$ vs k) and the dissipation rate of turbulence.

An interesting result is that this dissipation rate at small scales, but larger than the ion inertial length, seem to be larger than the random sweeping and eddy turnover time.

The manuscript in of interest. The authors need to improve the writing a little.

Particular suggestions (in [])

[Figure]

Abstract:

... yielding [] the result that the decay rate on spatial scales of about 1000 km (about 10 times larger than the ion inertial length) [which] is higher ...

... of plasma physical (and not fluid mechanical) [processes].

Introduction

... [A] Fourier representation of the two-time[] two-point...

... [Our] experimental access ...

... [For] the goal of measuring the ...

... Taylor$\sim$s frozen-in hypothesis in the spectral domain [(Taylor 1938)].

Breit-Wigner spectrum

... One may extend the frequencies from the real numbers (as oscillatory part)[] to the complex numbers ...

... rate $\sim$) [is] ...

... frequency $\sim 0$ is given as f BW = 2/\pi\gamma[, so that the] half-value of the peak is and [the] half-value width is realized when the decay rate $\sim$ satisfies the condition

Streamwise wavenumber-frequency spectrum

... of [xxx] and a frequency of [xxx]. The spectral extension is almost linear, and the slope of the extension (the propagation speed estimated [] from the ratio of the frequency to the wavenumber) roughly agrees with the mean flow speed, [xxx] km/s ...

... broadening is a function [of wavenumber].

Please explain better how the stream wavenumber-frequency spectrum is constructed. In particular, how many spacecrafts are used? What is the time resolution of the data? etc. This would make the manuscript more self-consistent. When constructing this

spectrum, is it possible to construct the stream wavenumber-frequency spectrum in normalized form with the local value of \Omega and V_A?

Fitting to the Breit-Wigner spectrum

... [A] fluctuation amplitude of 0.2

... lifetime of the fluctuation components is shorter [than that of] fluid turbulence ...

What are the meaning of these 2 sentences:

Interpretation as fluid picture. Doppler shift and Breit-Wigner type broadening rough estimate of mapping quality of Taylor~s hypothesis

Some missing sentences?

Please put the fitting errors in Fig. 6 and in the comparison with particular wave modes (whistlers or kinetic Alfven) in the previous reply. I believe that a nominal value for \Omega and V_A was used? How much do they vary in this interval. A factor of 2?

What is the value of the proton Larmor gyroradius. Can it be placed in Fig. 6? Does it vary much?

Discussion and Outlook

... the invariance of the spectral index is not [yet] guaranteed ...

---

## Author Response (AR1)

**Reply to referee comments**

I thank the both referees for their time for working on my manuscript with helpful suggestions. The comments raised by the referee 1 are already included and posted onto the discussion forum (with a kind answer from the referee, recommending the revision for publication). In the following, I reply to the referee 2 comments and add the communications with the referee 1 at the bottom.
* * *
**Referee 2**

- *The authors present an interesting method, based on the Breit-Wigner model for non Gaussian frequency broadening, to look at the stream wavenumber frequency spectrum constructed from cluster measurements in the solar wind and try estimate a dispersion relation ($\omega$ vs k) and the dissipation rate of turbulence.*

  *An interesting result is that this dissipation rate at small scales, but larger than the ion inertial length, seem to be larger than the random sweeping and eddy turnover time.*

  *The manuscript in of interest. The authors need to improve the writing a little.*

  *Particular suggestions (in [ ])*

  *Abstract:*

  - *... yielding [ ] the result that the decay rate on spatial scales of about 1000 km (about 10 times larger than the ion inertial length) [which] is higher ...*

    **Reply**: Done. (page 1, lines 4–5)

  - *... of plasma physical (and not fluid mechanical) [processes].*

    **Reply**: Done. (page 1, line 7)

  *Introduction*

  - *... [A] Fourier representation of the two-time[] two-point...*

    **Reply**: Done. (page 1, line 11)

  - *... [Our] experimental access ...*

    **Reply**: Done. (page 1, line 12)

  - *... [For] the goal of measuring the ...*

    **Reply**: Done. (page 2, line 8)

– ... *Taylor~s frozen-in hypothesis in the spectral domain [(Taylor 1938)].*

**Reply**: Done. It was due to "\cite" command inthe natbib package. I use "\citep" now. (page 2, line 10)

*Breit-Wigner spectrum*

– ... *One may extend the frequencies from the real numbers (as oscillatory part)[] to the complex numbers ...*

**Reply**: Done. (page 2, line 21)

– ... *rate ~) [is] ...*

**Reply**: Done. (page 3, line 6)

– ... *frequency ~ 0 is given as f BW = 2/\pi\gamma[, so that the] half-value of the peak is and [the] half-value width is realized when the decay rate ~ satisfies the condition*

**Reply**: Done. (page 3, lines 19–20)

*Streamwise wavenumber-frequency spectrum*

– ... *of [xxx] and a frequency of [xxx]. The spectral extension is almost linear, and the slope of the extension (the propagation speed estimated [] from the ratio of the frequency to the wavenumber) roughly agrees with the mean flow speed, [xxx] km/s ...*

**Reply**: Done. (page 5, lines 3–6 )

– ... *broadening is a function [of wavenumber].*

*Please explain better how the stream wavenumber-frequency spectrum is constructed. In particular, how many spacecrafts are used? What is the time resolution of the data? etc. This would make the manuscript more self-consistent. When constructing this spectrum, is it possible to construct the stream wavenumber-frequency spectrum in normalized form with the local value of \Omega and V_A?*

**Reply**: The answer was added as a paragraph starting as "The streamwise wavenumber-frequency spectrum is..." (page 5, line 10). Figure 4 shows both the unnormalized and the normalized spectra.
Reference to Glassmeier et al. (2001) and Motschmann et al. (1996) have been added.

*Fitting to the Breit-Wigner spectrum*

– ... [A] fluctuation amplitude of 0.2

**Reply**: Done. (page 7, line 6)

– ... *lifetime of the fluctuation components is shorter [than that of] fluid turbulence ...*

*What are the meaning of these 2 sentences: Interpretation as fluid picture. Doppler shift and Breit-Wigner type broadening rough estimate of mapping quality of Taylor~s hypothesis*

*Some missing sentences?*

**Reply**: The two sentences were my memorandum during the manuscript preparation. These phrases are deleted in the revision. The part of "than that of" has been corrected. (page 8, line 5)

*Please put the fitting errors in Fig. 6 and in the comparison with particular wave modes (whistlers or kinetic Alfven) in the previous reply. I believe that a nominal value for \Omega and V_A was used? How much do they vary in this interval. A factor of 2?*

**Reply**: The fitting errors are addded to the frequency error bar and the decay rate error bar in Fig. 6. The fitting error is the largest only at the smallest wavenumbers. The frequency error in the plasma rest frame mostly comes from the uncertainty in the flow velocity measurement, and I can show only the upper limit in the logarithmic plot style. The dispersion relations for the whistler mode and the kinetic Alfvén mode are computed for the measured local mean field values and a nominal propagation angle to the mean magnetic field. Changed places are:

  * "The frequency error in the plasma rest frame mostly comes from the uncertainty in the flow velocity measurement, and only upper limits can be shown in the logarithmic plot style." (page 9, lines 6–7)
  * "The fitting errors are addded in Fig. 6 as the frequency error bar and the decay rate error bar. The fitting error is the largest only at the smallest wavenumbers." (page 8, lines 7–9)
  * "The dispersion relations for the whistler mode and the kinetic Alfvén mode are computed for the measured local mean field values." (page 8, lines 17–18)

*What is the value of the proton Larmor gyroradius. Can it be placed in Fig. 6? Does it vary much?*

**Reply**: The gyro-radius for the thermal protons is estimated as about 84.6 km rad$^{-1}$. (page 4, lines 18–19) The wavenumber for the gyro-radius is $k_{\mathrm{gi}} = 1.18 \times 10^{-2}$ rad km$^{-1}$, and is beyond the plot range in Fig. 6. (page 8, lines 9–10)

*Discussion and Outlook*

– *... the invariance of the spectral index is not [yet] guaranteed ...*

**Reply**: Done. (page 10, line 9)
* * *
**Referee 1**

- *In this innovative paper the authors introduce the Breit-Wigner spectral distribution function in order to fit the observed data and extract the underlying turbulence decay rate. I believe that this is a worthwhile approach that should be made aware to the space physics community in general, since this method may be useful for unraveling some underlying physical processes. I have some questions, which I suggest the authors to consider in order to clarify the model.*

  *(Question) From the comparative plots of spectral distributions in Figure 1, Breit-Wigner distribution appears to have a superficial similarity with the kappa distribution. It seems to be that one may alternatively use the kappa spectral distribution to fit the data in Figure 5. So, the question is, why should one prefer BW distribution over the kappa or Lorentzian spectral function? Is there a rationale for choosing Breit-Wigner distribution over other models?*

  – Thank you very much for the positive evaluation. Underlying physical models are different between the kappa distribution and the Breit-Wigner distribution. The use of kappa distribution is developed for describing non-extensive statistical mechanics (e.g., non-extensive entropy), and is developed for discrete particles that have long-range interactions and correlations. Its application to the energy spectra for continuous turbulent fields is still in question, in particular, in interpreting the control parameter kappa in the distribution . On the other hand, the use of a Breit-Wigner (or Lorentzian) distribution has a solid background with a physical model in that the decay rate (appearing as an imaginary part of the frequency) can be measured experimentally and then immediately compared to wave or turbulence models.

    We added a paragraph on the above reply comment ("It is worthwhile to note ...") at the end of section 1 after the Breit-Wigner distribution is first introduced (page 2, lines 11–19).

- *(Question) Please specify what xxxs are in the following: Lines 19-20: wavenumber of xxx and a frequency of xxx? Line 21: mean flow speed, xxx km/s*

  – Oops. Done. "a wavenumber of $0.002 \, \mathrm{rad \, km^{-1}}$ and an angular frequency of about $0.75 \, \mathrm{rad \, s^{-1}}$, indicating a phase speed of about $375 \, \mathrm{km \, s^{-1}}$. (page 5, lines 3–4) and "$373 \, \mathrm{km \, s^{-1}}$" (page 5, lines 3–6).

- *(Question) In Figure 4, can the angular frequency versus wave number plot be fitted with some known dispersion relations? Fast, slow, or Alfven mode?*

  - Done. We corrected for the Doppler shift and compared the rest-frame frequencies with the obliquely-propagating whistler mode and the kinetic Alfvén mode. The whistler mode seems to be the best candidate to explain the measured frequencies, but the kinetic Alfvén mode can still remain a like candidate within the error bar. We added a subsection "Dispersion relation" on pages 8–9 and a figure of the dispersion relations (figure 7). Correspondingly, sentences are added in section 4 (page 10, lines 10—15) and rerefences to Gary (1993), Hollweg (1999), Lysak and Lotko (1996), Lysak (2008), and Perschke et al. (2014) are added.

Concluding comments by the referee 1

*The revisions make the MS much more clear in terms of motivation and presentation. In particular, I find it entirely convincing that the kappa model for the particles has a physical rationale based upon non-extensive statistical mechanics of long-range interaction and correlation. In contrast, Breit-Wigner distribution for fluctuations has another physical rationale, namely, lifetime of the fluctuations. Second, I find that the compari- son of fluctuation spectrum with dispersion relations is quite convincing, and it shows that the method is very useful. I thus recommend the publication in its present form.*